# An Initial Report of the Antimicrobial Activities of Volatiles Produced during Rapid Volatilization of Oils

**DOI:** 10.3390/antibiotics11121742

**Published:** 2022-12-02

**Authors:** Sangeetha Ramamurthy, Jonathan Kopel, David Westenberg, Shubhender Kapila

**Affiliations:** 1Department of Microbiology, Missouri University of Science and Technology, Rolla, MO 65401, USA; 2School of Medicine, Texas Tech University Health Sciences Center, Lubbock, TX 79430, USA; 3Department of Chemistry, Missouri University of Science and Technology, Rolla, MO 65401, USA

**Keywords:** antimicrobials, disinfectants, vapors, fog oil, methyl soyate, bacteria, viruses

## Abstract

Aerosols generated through volatilization and subsequent recondensation of oil vapors have been used as obscurant (smoke) screens during military operations since the early twentieth century. Specifically, a petroleum middle distillate known as the fog oil (FO) has been used in US military battlefields to create obscurant smoke screens. During a study on the feasibility of replacing petroleum-derived FO with vegetable oil-derived esters such as methyl soyate (MS), it was observed that that FO and MS aerosols and vapors did not exhibit detectable mutagenic activity but were lethal to Ames strains bacteria even after very short exposure periods. These results opened the potential use of oil-derived vapors as antimicrobial agents. Subsequent studies showed that optimal aerosol/vapor production conditions could further enhance disinfectant efficiency. For this purpose, we examined the antimicrobial activities of mineral oils and biogenic oil ester aerosols/vapors against a wide range of Gram-positive and Gram-negative bacteria. The results of the study showed that the aerosols/vapors obtained from mineral oil or vegetable oil ester under proper conditions can serve as an excellent antibacterial disinfectant.

## 1. Introduction

A disinfectant is an agent that destroys disease-causing microorganisms. Typically, disinfectants are substances applied to inanimate objects which distinguish them from antiseptics which are applied to living tissues. Some common examples of disinfectants are phenolic compounds, alcohols, quaternary ammonium compounds, chlorine compounds, aldehydes, and gases such as ethylene oxide. Disinfectants are used extensively in hospitals and other health care facilities for killing microorganisms on a variety of hard surfaces [1]. These substances are essential components of infection control practices and aid in the prevention of nosocomial infections. Mounting concerns over the potential for microbial contamination and infection risks in the food and general consumer markets have also led to increased use of antiseptics and disinfectants by the general public. Disinfectants comprise a wide variety of active chemical agents (or “biocides”). Many of these compounds have been used for hundreds of years for antisepsis, disinfection, and preservation. Despite this, the modes of action of active agents remain largely unknown. In general, biocides have a broader spectrum of activity than antibiotics, and, while antibiotics tend to have specific intracellular targets, biocides may have multiple targets. One example of such biocides includes aerosols, such as fog and mineral oils.

Aerosols generated through volatilization and subsequent recondensation of oil vapors have been used as obscurant (smoke) screens during military operations since the early twentieth century [2,3]. The battlefield obscurant used for impairing view in the visible region produces a dense white cloud which attenuates light transmission in the visible region of the electromagnetic spectrum thereby confounding enemy sensors and smart munitions [2,3]. For the past four decades, fog oil (de-aromatized middle distillate petroleum (MIL-F-12070E)) has been used by the US military as obscurant in the battlefield and during training exercises. Recently, use of biogenic (vegetable) oils has been investigated as a substitute for the fog oil [4,5,6]. Biogenic oils are non-petroleum-based oils with physical properties similar to fog oil. Compared to other disinfectants, fog and biogenic oils have properties that make them superior to other commonly used disinfectants [7,8,9,10,11,12,13]. First, fog oils are free of potentially carcinogenic poly aromatic hydrocarbons (PAHs). Second, the individual components of these oils (e.g., esters, aldehydes, terpenes, alcohols, and hydrocarbons) are more effective antimicrobial agents when used in the vapor phase than in the solution phase [6]. Lastly, the diversity of compounds allows for a broad range and potency of antimicrobial activities against different bacteria [6]. The antimicrobial activity of oils is associated with their physico-chemical properties, which affect the depth of penetration into the bilayer. For example, lipophilic oils can lower the plasma membrane integrity in mammalian and bacterial cells.

During these evaluations, the fog oil and vegetable oil esters and their vapors were subjected to the Ames test [14]. The Ames test is one of the most widely used assay techniques to test the mutagenicity of chemicals [14]. The Ames test involves exposure of auxotrophic strains of *Salmonella typhimurium* (histidine requiring) to potential mutagenic compounds and measuring the relative increase in reversion of histidine auxotrophy [14]. The antibacterial activity of the fog oil and vegetable esters was observed through the elimination of the background growth in the Ames test. The results of the Ames test revealed that the vapors obtained from oils under certain conditions were not mutagenic but lethal to bacterial strains used in the Ames test. These results opened the potential use of oil-derived vapors as antimicrobial agents. The overall objective of this study was to evaluate antimicrobial properties of volatiles produced during rapid volatilization of biogenic oil esters and mineral oils through modifications in the lethality of the biogenic oil esters through different times, temperatures, and exposures.

## 2. Results

### 2.1. Mutagenicity of Aerosols and Vapors Obtained through Rapid Volatilization of MS, LOME, and FO

The Ames test has often been used to test mutagenicity of chemicals. According to the test protocol of the assay, a test chemical is said to be a potential mutagen when its presence leads to a 2–3-fold increase in the number of revertants over the carrier solvent control. The results of the aerosol and vapor exposure in terms of revertant colony counts are shown graphically in Figure 1 and Figure 2. The numbers of colonies on plates exposed to oil aerosols and vapors for short periods (<120 s) were nearly the same as for the control plates left in clean air within in the exposure chamber. Longer exposures (>120 s) to aerosols and vapors or vapors alone resulted in a dramatic decrease in the number of colonies on the plates. Exposures longer than 300 s resulted in complete disappearance of bacterial colonies, indicating that oil aerosols and/or vapors are toxic to the *Salmonella* stain TA97. Even after very short exposures (<120 s), only pin-point colonies were observed in some of the samples. These might arise since high-level toxicity resulted in more histidine being available to the surviving His^-^ bacteria on a per cell basis.

### 2.2. Toxicity of Oil Aerosols and Vapors towards Ames Strains

Ames assay results showed that fog oil and methyl ester aerosols were lethal to Ames strains. More comprehensive experiments were carried out to assess the antibacterial activity of the oil aerosols and vapors. In these experiments, selected test microorganisms were exposed to fog oil, methyl soyate, soybean oil isopropyl ester, and olive oil methyl ester aerosols and vapors. Results of the tests are presented in Table 1 and Table 2. The results showed that the antibacterial activity of vegetable oil esters was higher than the antibacterial activity exhibited by fog oil. This could be related to the presence of unsaturated fatty acids that can readily give rise to aldehydes and ketones. The highest antimicrobial activity was shown by methyl soyate and olive oil methyl ester aerosol–vapors.

### 2.3. Permeability of Antibacterial Component(s) through Paper

A set of experiments were carried out to discern whether the antibacterial activity was related to the aerosols or the vapors. In these experiments, nutrient agar plates were inoculated with selected bacterial species and exposed to methyl soyate aerosols-vapors or the vapor filtered through paper barriers. The paper barriers consisted of a thin Kimwipe^®^ paper with a hole, an intact paper, or a 1 mm thick paper. The paper barriers were placed on top of the nutrient agar plates inoculated with selected bacterial species. Exposures were carried out for different time periods. The control plates were exposed only to clean air. Results obtained with plates covered with a Kimwipe^®^ paper (with a 6 mm hole) are summarized in Table 3. No bacterial growth was observed on plates which were exposed to either fog oil or methyl soyate for more than five minutes. None of the *Salmonella* sp. survived the ten-minute exposure. In most cases, a 2 min exposure to methyl soyate vapors was sufficient to kill most strains. Little or no oil residue was observed on the exposed plate. In another set of experiments, nutrient agar plates inoculated with *Salmonella* sp. were covered with a Kimwipe^®^. Results of the experiments are summarized in Table 4. Results were identical to those obtained with Kimwipe^®^ (with a 6 mm hole)-covered plates inoculated with Salmonella. The active antibacterial components were even able to permeate through a 1mm paper with a 6 mm opening in the middle or a “non-porous” paper without the opening. Results of the experiments are summarized in Table 2, Table 5, and Table 6, respectively.

### 2.4. Retention of Antibacterial Agents by Nutrient Agar

Results presented in the preceding section clearly show that the antibacterial components of the oil aerosol/vapor are not removed through filtration through paper, indicating that the antibacterial activity is not related to oil aerosols but stems from the volatile components present in the vapor phase. It was observed that the exposure of nutrient agar to aerosol/vapor exposure introduces antimicrobial substances in the agar media, making it unsuitable for bacterial growth. If the toxic substance had a high vapor pressure and low solubility in the agar media, these components should have been lost in the vapor phase when media were placed in the well-ventilated sterile hood. The loss would make the medium fit for the growth of *Salmonella* strains. An experiment was conducted during which the nutrient agar plates were exposed to aerosol/vapor for 5 min. The exposed plates were then placed in a sterile hood for 5, 10, and 30 min. The plates were then inoculated with bacterial cultures and incubated to grow. The plates were examined after 24 h for bacterial growth. The results obtained from the experiment are summarized in Table 7. The results indicate that toxicity induced by exposure to methyl soyate aerosols can be reversed by allowing sufficient time for the volatilization to occur. The results with fog oil indicate that the antibacterial activity is retained even after 30 min of volatilization time.

### 2.5. Standard Test for Surface Disinfectants

The efficacy of oil aerosol/vapor for disinfecting solid surfaces was tested through a test involving pins coated with bacterial cultures. The results of the experiment are summarized in Table 8. The pins contaminated with bacterial cultures were exposed to aerosols of fog oil and methyl soyate as well as neat oils. The results indicate that in comparison to neat oil, the aerosols of fog oil and methyl soyate seem to be more effective against the bacteria for a 2 min aerosol exposure period. The results shows that the aerosols generated from the oils can act as a surface disinfectant.

### 2.6. Oil Aerosols for Surface Sterilization

Oil aerosols appear to be effective surface disinfectants. To determine if the aerosols can sterilize surfaces, it must be determined if the aerosols can kill bacterial endospores. Table 9 and Table 10 show the results of aerosol exposure of bacterial endospores to fog oil and methyl soyate aerosols. Higher temperatures than those used in previous experiments were used because it was determined that higher temperatures were more effective at killing a broader spectrum of microorganisms (data not shown). The results indicate that fog oil aerosols were successful in killing *Bacillus* endospores after 2 h of exposure but methyl soyate aerosols were not effective against endospores of *Bacillus*.

### 2.7. Oil Aerosols were Lethal to Bacterial Species and Lethality Was Temperature Dependent

Table 11 shows test results of exposure of other bacteria to oil aerosols. The result from preliminary experiments shows that oil aerosols were lethal to Ames strains and more tests were performed in the laboratory to determine the antibacterial efficiency of oil aerosols against other bacteria. It was observed that exposure to oil aerosols was lethal to other bacteria and oil aerosols generated at different temperatures showed a difference in killing efficiency.

### 2.8. Relative Antimicrobial Activity of Methyl Soyate vs. Other Oils

The antibacterial efficiency of other petroleum-based oils such as JP-8 and diesel is being compared with methyl soyate. Since the idea is to replace petroleum-based oils with vegetable oils such as methyl soyate to create smoke screens in military training exercises, a series of experiments was carried out to document the fact that methyl soyate is more effective when compared to diesel and JP-8 rather than fog oil. Results indicate that diesel aerosols were lethal to all bacteria only when generated at temperatures as high as 650 °C. This is as effective as methyl soyate whereas we previously demonstrated that methyl soyate is effective for a 5 min exposure period even at temperatures such as 550 °C. JP-8 aerosols when tested at temperatures as high as 650 °C was not effective with a 5 min exposure period of bacteria. This could also be attributed to the fact that diesel and JP-8 are made up of more saturated hydrocarbons as compared to methyl soyate which contains more unsaturated linoleate and oleate content which produces more volatile antibacterial compounds on vaporization and condensation. Table 12 shows results of aerosol exposure of methyl soyate, diesel, and JP-8 generated at 650 °C.

## 3. Discussion

Preliminary experiments with aerosols generated from fog oil and methyl soyate were lethal to Ames strains. Subsequent experiments showed that components in the aerosols could readily diffuse through barriers such as Kimwipe^®^ and stationery paper and be lethal to the bacterial lawn on the surface of the medium. It was also found that this lethality was temperature and time dependent. Further experiments showed that the lethality of the aerosols was not restricted only to Ames strains but also to other Gram-positive and Gram-negative bacteria and spore formers such as *Bacillus subtilis*. Future studies should focus on optimizing conditions required to increase the antifungal efficiency. Gas chromatography–mass spectroscopy studies have shown that these aerosols consist mainly of long chain aldehydes, ketones, and fatty acids. Further experiments should be performed with neat compounds and with similar concentrations as found in the vapor. Although the mechanism through which biogenic oils eliminate bacteria is unknown, we hypothesize that the primary mode of action is the destruction of the bacterial cell wall and membrane. However, we suspect that there could be other mechanisms by which these biogenic oils eliminate bacteria. The possibility of using this disinfectant technology to disinfect contaminated water sites should also be explored further. With the obtained data, we believe that there are a variety of applications to use biogenic oils within a clinical environment. To examine this, it would be beneficial to assess the application of biogenic oils by changing the delivery mechanism and methods for concentrating these oil products. Future studies should also focus on the exact mechanism of action of the vapor disinfectant on bacteria. Overall, our study demonstrated the potential use of volatile biogenic oils as novel antimicrobials and disinfectants for future investigation and clinical application.

## 4. Methods

Oils: Methyl soyate was purchased from AG Environmental Products LLC (Lenexa, Kansas). Fog oil was obtained from US Army Chemical School (Fort Leonard Wood, MO, USA). Fog oil is a middle distillate of petroleum similar to the commercially available mineral oil. It is used for generation of smoke screens through vaporization and condensation processes. Soy oil propyl ester (SOPE), olive oil methyl ester (OOME), and linseed oil methyl esters were prepared in the laboratory through transesterification reactions of the oils with the appropriate alcohol in the presence of a basic catalyst. The biogenic oil esters used had different carbon chain lengths and unsaturation. Diesel is a specific fractional distillate of fuel oil (mostly petroleum) and made up of 75% saturated hydrocarbons and 25% aromatic hydrocarbons. It was obtained from a local gas station (Rolla, MO, USA). Jet Propellant 8 (JP-8) is a kerosene-like fuel for jet aircrafts and Department of Defense (DOD) tactical vehicles. It was obtained from US Army Chemical School at Fort Leonard Wood, MO, USA.

Ames strains: TA97, TA98, and TA100 were obtained from Dr. Bruce Ames Laboratory, University of California, Berkeley [14]. These stains are histidine-dependent *Salmonella* strains each carrying different mutations in various genes in the histidine operon. Stock cultures were stored at −80 °C and fresh sub-cultures from frozen stock were prepared on a regular basis [14]. Active cultures were transferred to fresh media every week.

Bacterial cultures: *K. pneumoniae*, *P. aeruginosa*, *E. cloacae*, *E. coli*, *S. marcescens*, *S. flexneri*, *B. megaterium*, *S. aureus*, and *S. typhimuirum* were obtained from American Type Collection Center (ATCC) and stock cultures were stored at −80 °C. Fresh sub-cultures from frozen stock were prepared on a regular basis. Active cultures were transferred to fresh media every week.

Bacterial culture media: The following culture media were used for basic bacterial culturing, mutagenicity testing, and exposure experiments. Tryptic soy agar is a nutrient medium suitable for sub-culturing, serial dilutions, and exposure experiments. It was prepared with 1.5% agar and 3.5% tryptic soy broth. Glucose minimal media is the bottom agar used for mutagenicity assay. It was prepared with 1.5% agar, 1X Vogel Bonner (VB) salt solution, and 0.2% glucose solution. Agar was added to water in a flask and autoclaved 10 X VB stock solutions and 10% glucose solution were added, mixed, and poured into Petri dishes. Top agar supplemented with histidine/biotin was used to deliver the bacteria, chemical, and buffer or S9 mix to the bottom agar. The main ingredients include 0.7% agar and sodium chloride. Filter-sterilized histidine/biotine was added after autoclaving.

S9 mix creation: Some carcinogenic chemicals are biologically inactive unless they are metabolized in the liver to active forms. Since bacteria do not have appropriate metabolic capability, an exogenous metabolic mammalian organ activation system can be added to the Petri dish together with the test chemical and the bacteria. Tests were performed with and without S9 to determine if liver metabolism significantly impacts mutagenicity. S9 was generated through a rodent metabolic activation system which mainly consists of 90008g supernatant fraction of a rat liver homogenate and was delivered to the test system in the presence of NADP and cofactor NAPDH-supported oxidation. Sodium phosphate buffer: This was added to the control sample (without S9) and used to test chemicals in the absence of metabolic activation. Monobasic sodium phosphates (0.1 M) and dibasic sodium phosphates (0.1 M) were combined to achieve a pH of 7.4. The buffer was autoclaved and stored in screw-cap bottles.

Exposure of Ames strains to aerosols: All smoke generation and exposure experiments were carried out in a fume hood. The exposure of Ames strains and other bacteria was conducted in the gas-tight stainless steel–borosilicate chamber described earlier. The chamber was placed inside the fume hood. The test specimens were placed inside or retrieved from the chamber through a hinged door. In the following experiments, the air flow was maintained at 10 L min^−1^ while oil flow was maintained at 0.5 mL min^−1^ unless otherwise mentioned. Temperature inside the chamber was 24 °C.

Mutagenicity testing of oil aerosols and vapor produced from linseed oil methyl ester, fog oil, and methyl soyate: Top agar was mixed with Ames strain TA97 and S9 mix or buffer. The contents of the tube were poured on the surface of minimal glucose agar medium. The plates were then exposed to oil aerosols and vapor obtained through rapid volatilization of linseed oil methyl ester, fog oil, and methyl soyate for specific exposure periods. The plates were incubated at 37 °C for 72 h. The plates were examined for revertant colonies.

Direct exposure to oil aerosols and vapor: Exposure of *Salmonella* strains TA97, TA98, and TA100 to oil aerosols and vapors of fog oil, methyl soyate, soybean oil isopropyl ester, and olive oil methyl ester was performed. A 5 mL culture (100 μL) was added to the Petri dish and spread with glass beads to produce a lawn of bacteria. The plates were exposed to oil aerosols and vapors for specific exposure periods, removed from the chamber, and incubated at 37 °C for 24 h.

Oil vs. vapor: Two sets of nutrient agar plates were used in this experiment. In one set, the plates were covered completely with a Kimwipe^®^ and stationery paper. In the second set, the plates were covered with papers with a hole in the center. After the exposure, the plates were transferred to a sterile hood. A 24 h culture was added as described above and the plates were incubated for 24 h at 37 °C.

Residual activity: Nutrient agar plates were exposed to fog oil and methyl soyate aerosols for 5 min. The plates were kept open in a sterile hood for 5 min, 10 min, and 30 min durations. A 24 h culture (100 μL) was used and the plates were incubated for 24 h at 37 °C. As a control, unexposed plates were incubated for 24 h at 37 °C.

Standard test for disinfectant properties: A test to examine the disinfectant efficacy on solid surfaces was also carried out. Sterile metal pins were inoculated with 24 h broth culture of TA97, 98, and 100, and dried in a sterile hood. Pins treated with each organism were exposed to oil aerosols and neat oils. Baby oil and pins not exposed to smoke were used as negative controls. The pins were exposed for 2 min and inoculated into 5 mL sterile nutrient broth and incubated for 24 h at 37 °C with shaking (200 rpm).

Exposure of bacterial endospores to aerosols: Duo-spore sterility test kits (obtained from Fisher Scientific Waltham, MA, USA), generally used to test the efficiency of autoclaves, were used to test the effectiveness of oil aerosols against bacterial endospores. The strips contain endospores of *Bacillus subtilis* and *Bacillus stearothermophilus* enclosed in a sterile envelope. The strips were exposed to fog oil and methyl soyate aerosols generated at 550 °C and 650 °C for exposure periods of 60 and 120 min. After exposure, the strips were inoculated in tryptic soy broths. The broths were incubated at 37 °C for 24 h with shaking (200 rpm).

Exposure of bacteria: A 24 h culture (100 μL) of *Klebsiella pneumoniae*, *Pseudomonas aeruginosa*, *Enterococcus cloacae*, *Escherichia coli*, and *Serratia marcescens* was added to Petri dishes with nutrient agar and spread using sterile glass beads. The plates were kept inside the glass chamber for a 30 min time period.

Methyl soyate vs. other oils: A 24 h culture (100 μL) of *Klebsiella pneumoniae*, *Pseudomonas aeruginosa*, *Enterococcus cloacae*, *Escherichia coli*, *Serratia marcescens*, *Staphylococcus aureus*, *Bacillus megaterium*, *Shigella flexneri*, and *Salmonella typhimurium* was cultured on nutrient agar plates. The plates were exposed to methyl soyate, JP-8, and diesel aerosols and vapors generated at 650 °C for 15 and 30 min exposure periods.

## Figures and Tables

**Figure 1 antibiotics-11-01742-f001:**
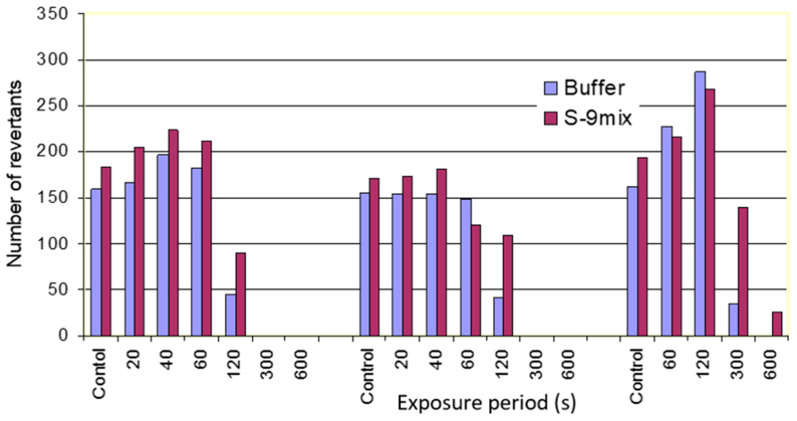
Number of TA97 strain revertant colonies observed after exposure to oil aerosols, indicating that the oil aerosols and vapors were not mutagenic to the Ames strains.

**Figure 2 antibiotics-11-01742-f002:**
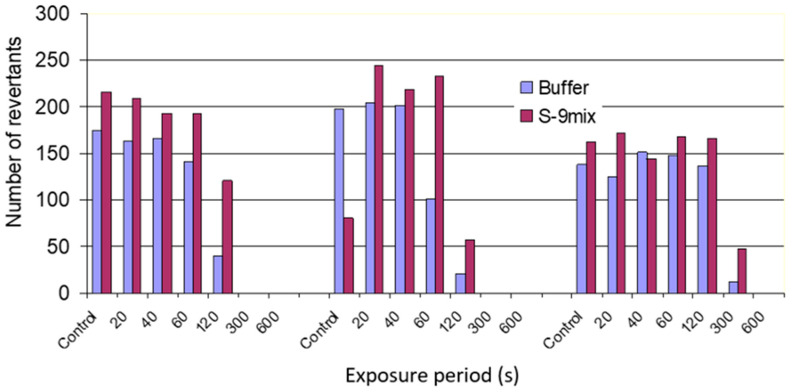
Number of TA97 strain revertant colonies observed after exposure to vapors.

**Table 1 antibiotics-11-01742-t001:** Relative antibacterial activities of vegetable oil esters and fog oil aerosols.

Methyl Ester/Oil	Methyl Soyate			Fog Oil			SOPE			OOME		
Exposure time (min)	2	5	10	2	5	10	2	5	10	2	5	10
Control	+	+	+	+	+	+	+	+	+	+	+	+
TA97	+	+	-	-	-	-	+	PG	-	-	-	-
TA98	-	-	-	PG	-	-	PG	-	-	-	-	-
TA100	PG	-	-	+	+	-	+	-	-	-	-	-

PG = Partial growth, + indicates growth, - indicates no growth.

**Table 2 antibiotics-11-01742-t002:** Relative antibacterial activities of vegetable oil esters and fog oil vapors.

Methyl Ester/Oil	Methyl Soyate			Fog Oil			SOPE			OOME		
Exposure time (min)	2	5	10	2	5	10	2	5	10	2	5	10
Control	+	+	+	+	+	+	+	+	+	+	+	+
TA97	+	-	-	+	-	-	PG	-	-	+	-	-
TA98	-	-	-	PG	+	-	-	-	-	-	-	-
TA100	PG	-	-	+	+	-	PG	-	-	+	-	-

PG indicates partial growth, + indicates growth, and - indicates no growth.

**Table 3 antibiotics-11-01742-t003:** Exposure results of plates covered with Kimwipe^®^.

Organism	Control	Fog Oil			Methyl Soyate		
		2 min	5 min	10 min	2 min	5 min	10 min
TA97	+	+	-	-	+	+	-
TA98	+	+	-	-	+	-	-
TA100	+	+	-	-	+	+	-

+ indicates growth, and - indicates no growth.

**Table 4 antibiotics-11-01742-t004:** Exposure results of plates covered with Kimwipe^®^ with a center hole.

Organism	Control	Fog Oil			Methyl Soyate		
		2 min	5 min	10 min	2 min	5 min	10 min
TA97	+	-	-	-	+	-	-
TA98	+	-	-	-	-	-	-
TA100	+	-	-	-	-	-	-

+ indicates growth, and - indicates no growth.

**Table 5 antibiotics-11-01742-t005:** Exposure results of plates covered with stationery paper.

Organism	Control	Fog Oil			Methyl Soyate		
		2 min	5 min	10 min	2 min	5 min	10 min
TA97	+	+	-	-	+	+	-
TA98	+	+	-	-	+	-	-
TA100	+	+	-	-	+	-	-

+ indicates growth, and - indicates no growth.

**Table 6 antibiotics-11-01742-t006:** Exposure results of plates covered with stationery paper with a hole in the center.

Organism	Control	Fog Oil			Methyl Soyate		
		2 min	5 min	10 min	2 min	5 min	10 min
TA97	+	+	-	-	+	+	-
TA98	+	+	-	-	+	-	-
TA100	+	+	-	-	+	-	-

+ indicates growth, and - indicates no growth.

**Table 7 antibiotics-11-01742-t007:** Recovery results following aerosol exposure.

Organism	Control	Fog Oil			Methyl Soyate		
		5 min	10 min	30 min	5 min	10 min	30 min
TA97	+	-	-	-	+	+	+
TA98	+	-	-	-	-	+	-
TA100	+	-	-	-	-	+	+

+ indicates growth, and - indicates no growth.

**Table 8 antibiotics-11-01742-t008:** Results of standard test for surface disinfectants.

	TA97	TA98	TA100
Control (for oil aerosol)	+	+	+
Control (for oil)	+	+	+
Fog oil aerosol	-	-	-
Fog oil (neat)	+	+	+
MS aerosol	-	-	-
MS (neat)	+	+	+

+ indicates growth, and - indicates no growth.

**Table 9 antibiotics-11-01742-t009:** Results of exposure of sporulating bacteria to fog oil aerosols.

Temp (^o^C)	550 °C		650 °C	
Duration	60 min	120 min	60 min	120 min
Control	+	+	+	+
Strips	+	-	+	-

+ indicates growth, and - indicates no growth.

**Table 10 antibiotics-11-01742-t010:** Results of exposure of sporulating bacteria to methyl soyate aerosols.

Temp (°C)	550 °C		650 °C	
Duration	60 min	120 min	60 min	120 min
Control	+	+	+	+
Strips	+	+	+	+

+ indicates growth, and - indicates no growth.

**Table 11 antibiotics-11-01742-t011:** Results of exposure of bacteria to oil aerosols.

		Fog Oil			Methyl Soyate		
		Temperature (°C)					
Bacteria	Control	350	450	550	350	450	550
*K. pneumoniae*	+	+	-	-	+	-	-
*P. aeruginosa*	+	+	-	-	+	-	-
*E. cloacae*	+	+	-	-	+	-	-
*E. coli*	+	+	-	-	+	+	-
*S. marcescens*	+	+	-	-	+	-	-

+ indicates growth, and - indicates no growth.

**Table 12 antibiotics-11-01742-t012:** Relative antimicrobial activity of oil aerosols produced from methyl soyate and other oils.

		Diesel		Methyl Soyate		JP-8	
Bacteria	Control	5 min	15 min	5 min	15 min	5 min	15 min
*S. flexneri*	+	-	-	-	-	PG	-
*K. pneumoniae*	+	-	-	-	-	+	-
*B. megaterium*	+	-	-	-	-	-	-
*P. aeruginosa*	+	-	-	-	-	+	-
*E. cloacae*	+	-	-	-	-	+	-
*S. aureus*	+	-	-	-	-	+	-
*E. coli*	+	-	-	+	-	+	-
*S. typhimuirum*	+	-	-	-	-	+	-
*S. marcescens*	+	-	-	-	-	-	-

+ indicates growth, and - indicates no growth.

## Data Availability

Not applicable.

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
