# Peer review of "An Initial Report of the Antimicrobial Activities of Volatiles Produced during Rapid Volatilization of Oils"

_antibiotics, 2022, doi:10.3390/antibiotics11121742_

Round 1

Reviewer 1 Report

Please revise your manuscript and take in consideration the commntes as reported in the attached pdf file

Author Response

Reviewer #1:

Comments and Suggestions for Authors - Please revise your manuscript and take in consideration the comments as reported in the attached pdf file

We thank the reviewer for their comments

It seems to be very general information with 11 references, try to specify into 2 or 3 phrases with references from which you have collected already.

We thank the reviewer for their comment. The number of citations was reduced in the manuscript for this sentence.

Highlighted Text/No Comment: “Bacterial cultures: Klebsiella pneumoniae, Pseudomonas aeruginosa, Enterococcus 64 cloacae, Escherichia coli, Serratia marcescens, Shigella flexneri, Bacillus megaterium, 65 Staphylococcus aureus, Salmonella typhimuirum”

We did not see a comment here. However, we made sure that the bacteria were abbreviated in this section of the methods.

Bacterial Culture Media - the subtitle does not include other information in this paragraph.

We thank the reviewer for their comment. The section on S-9 Mix Creation was not bolded. We added this to the methods to separate what was bacterial culture media and the specifics related to the creation of the S-9 Mix.

A 5 hr culture (100 μL) – can you check if this is correct?

We appreciate the reviewer’s comment. We meant A 5 ml culture (100 μL)

Results - Are the results should be after M&M or before, please check the regulations of the journal?

The methods were moved to the correct location after the discussion.

Table 1 – For each kind, please make only one table cell. eliminate the lines under each kind

We thank the reviewer for their comment. We made sure to eliminate the separate lines and make each major group one section.

Is the table 2 similar to Fig 1, please check?

We thank the reviewer to their comment. We decided for clarity to remove table 1 and 2 and, instead, keep Figures 1 and 2 in the manuscript. Figure 1 goes with Table 1; Figure 2 goes with Table 2. We apologize for the confusion. In addition, Table 2 is not similar to Figure 1. Figure 1 and Figure 2 were focusing on different treatments. The Tables were merely showing the raw data in each treatment group.

Reviewer 2 Report

Dear Author, I reviewed the manuscript (antibiotics-2001348) entitled Evaluation of antimicrobial activities of volatiles produced during rapid volatilization of oils. This manuscript presents relevant information about the antibacterial potential of minor bioactive components of fog oils. However, some sections of the presented data can be improved. For this reason, I consider that this manuscript needs minor changes to be considered for publication in this journal. 

Additional comments.

Highlight the advantages of using fog oils as antimicrobial compounds.

Check the paragraph extension in this manuscript.

Try to indicate how the antibacterial activity was detected. 

Include an experimental design containing statistical factors and response variables in the statistical analyses applied to the findings of this research.

Include a possible mode of action of bioactive compounds against pathogenic bacteria in the tested fog oils.

Try to compare the obtained findings with similar assays where the antibacterial potential of fog oils was evaluated. 

Try to include a statistical description in the figures that require it. 

Include future trends to keep working with the obtained data. 

Try to conclude with a general statement of the most relevant part of this study.

Author Response

Reviewer #2:

Dear Author, I reviewed the manuscript (antibiotics-2001348) entitled Evaluation of antimicrobial activities of volatiles produced during rapid volatilization of oils. This manuscript presents relevant information about the antibacterial potential of minor bioactive components of fog oils. However, some sections of the presented data can be improved. For this reason, I consider that this manuscript needs minor changes to be considered for publication in this journal. 

We thank the reviewer for taking time to review the manuscript.

Highlight the advantages of using fog oils as antimicrobial compounds.

We wrote the following to describe the advantages of fog oils: “Biogenic oils are non-petroleum-based oils with physical properties similar to fog oil. Compared to other disinfectants, fog and biogenic oils have properties that make them superior to other commonly used disinfectants. First, FOG oils are free of potentially carcinogenic poly aromatic hydrocarbons (PAHs). Second, the individual components of these oils (e.g. esters, aldehydes, terpenes, alcohols and hydrocarbons) are more effective antimicrobial agents when used in the vapor phase than in the solution phase.5 Lastly, the diversity of compounds allows for a broad range and potency of antimicrobial activities against different bacteria.5 The antimicrobial activity of oils is associated with their physic-chemical properties, which affects the depth of penetration into the bilayer. For example, lipophilic oils can lower the plasma membrane integrity in mammalian and bacterial cells”.

Check the paragraph extension in this manuscript.

We appreciate the reviewer’s comment. We standardized the paragraph extension

Try to indicate how the antibacterial activity was detected. 

We appreciate the reviewer’s comment. We detected antibacterial activity through our observation that the background growth of the Ames test was eliminated in the presence of these volatile oils.

Include an experimental design containing statistical factors and response variables in the statistical analyses applied to the findings of this research.

We appreciate the reviewer’s comment. This is more of qualitative study to examine biogenic oils and their antimicrobial activity. Figures 1 and 2 was not to examine the superiority of vapors and biogenic oils. We wanted to demonstrate the ability to see if biogenic oils were a viable alternative to petroleum oils in eliminating bacteria as a qualitive study.

Include a possible mode of action of bioactive compounds against pathogenic bacteria in the tested fog oils.

We appreciate the reviewer’s comments. This is an active area of investigation still. The mechanism for fog oil’s antibacterial activity would be good for a future study. However, we hypothesize that the primary mode of action is the destruction of the bacterial cell wall and membrane. However, we suspect that there could be other mechanisms by which these fog oils eliminate bacteria.

Try to compare the obtained findings with similar assays where the antibacterial potential of fog oils was evaluated. 

We appreciate the reviewer’s comments. We checked pubmed and other literature sources. We could not find similar antibacterial studies on fog oils.

Try to include a statistical description in the figures that require it. 

We appreciate the reviewer’s comment. As mentioned previously, our study was more qualitative. Figures 1 and 2 were comparing incremental exposure times independently, which were to guide our future investigations later in the paper. However, we plan to investigate this effect through more thorough quantitative analysis.

Include future trends to keep working with the obtained data. 

We appreciate the reviewer’s comments. With the obtained data, we believe that there a variety of applications using biogenic oils within a clinical environment. To examine this, it would be beneficial to assess the application of biogenic oils by changing the delivery mechanism and methods for concentrating these oil products.  

Try to conclude with a general statement of the most relevant part of this study.

We appreciate the reviewer’s comment. We wrote the following in the paper, “Overall, our study demonstrated the potential use of volatile biogenic oils as novel antimicrobials and disinfectants for future investigation and clinical application.”

Reviewer 3 Report

The manuscript “Evaluation of antimicrobial activities of volatiles produced during rapid volatilization of oils” presents an interesting proposal when describing the results of the study that shows that aerosols/vapors obtained from mineral oil and vegetable oil may have relevant antimicrobial properties.   However, the manuscript needs adaptations before being published. - In the bacterial cultures item of the methodology, the names of the microorganisms are not in italics. - Replace the term microbes with microorganisms throughout the text. - No item method has no used methodology reference. Scientific articles the experiments were in which articles? - No items Methyl Soy vs. other line 138 oils as the viability of microorganisms with this temperature tested? - In tables 1 and 2 in the column the exposure agent defines one at the foot of the table. - Line 252 Oil aerosols for surface sterilization. Clarify that tests were performed to find out if a Bacillus bacterium tested sporulated? - Lack of statistical tests to assess and evaluate the antibacterial activity of an oil was significantly superior to another tested. - The first paragraph of the discussion item should be moved to the introduction as it does not discuss the results found. This item needs to be improved with a discussion based on the results found and based on the scientific literature. - In the discussion and conclusion, it is stated that the lethality of aerosols may be related to long-chain aldehydes, ketones and fats. However, at no time was any chemical analysis or study performed to substantiate these claims. - I suggest a search to cite new studies, most of the articles mentioned in the reference have more than ten years of publication.

Author Response

Reviewer #3:

The manuscript “Evaluation of antimicrobial activities of volatiles produced during rapid volatilization of oils” presents an interesting proposal when describing the results of the study that shows that aerosols/vapors obtained from mineral oil and vegetable oil may have relevant antimicrobial properties.  However, the manuscript needs adaptations before being published. 

- In the bacterial cultures item of the methodology, the names of the microorganisms are not in italics. 

We appreciate the reviewer’s comment. The names of the microorganisms were put in italics.

- Replace the term microbes with microorganisms throughout the text.

We thank the reviewer for their comment. We changed all instances of microbes to microorganisms

 - No item method has no used methodology reference. Scientific articles the experiments were in which articles? 

We appreciate the reviewer’s comment. We added a reference for the Ames test. The rest of the methodology was novel and developed in our lab to assess the volatility of these biogenic oils.

- No items Methyl Soy vs. other line 138 oils as the viability of microorganisms with this temperature tested? 

We appreciate the reviewer’s comment. We used this temperature based on the previous data shown in the manuscript that the 650 degree Celsius had the greatest antimicrobial effect for the volatile biogenic oils.

- In tables 1 and 2 in the column the exposure agent defines one at the foot of the table. 

We appreciate the reviewer’s comment. As mentioned for reviewer 1, we removed tables 1 and 2 from the manuscript.

- Line 252 Oil aerosols for surface sterilization. Clarify that tests were performed to find out if a Bacillus bacterium tested sporulated?

We appreciate the reviewer’s comment. We assessed whether the biogenic oils could eliminate spores. However, future studies could examine whether these oils can prevent bacterial spores from sporulating.

 - Lack of statistical tests to assess and evaluate the antibacterial activity of an oil was significantly superior to another tested. 

We appreciate the reviewer’s comment. Figures 1 and 2 was not to examine the superiority of vapors and biogenic oils. We wanted to demonstrate the ability to see if biogenic oils were a viable alternative to petroleum oils in eliminating bacteria.

- The first paragraph of the discussion item should be moved to the introduction as it does not discuss the results found. This item needs to be improved with a discussion based on the results found and based on the scientific literature. 

We appreciate the reviewer’s comment. We made the changes to the organization of the manuscript.

- In the discussion and conclusion, it is stated that the lethality of aerosols may be related to long-chain aldehydes, ketones and fats. However, at no time was any chemical analysis or study performed to substantiate these claims. 

We appreciate the reviewer’s comment. Although this was not mentioned in this manuscript, other studies, such as the one listed below, mentions the effect of chemical composition and chain length has on the elimination of different infections.

He, C.-N.; Ye, W.-Q.; Zhu, Y.-Y.; Zhou, W.-W. Antifungal Activity of Volatile Organic Compounds Produced by Bacillus methylotrophicus and Bacillus thuringiensis against Five Common Spoilage Fungi on Loquats. Molecules 202025, 3360. https://doi.org/10.3390/molecules25153360

- I suggest a search to cite new studies, most of the articles mentioned in the reference have more than ten years of publication.

We appreciate the reviewer’s comment. We added some more recent publications to this manuscript. These are shown in citations 7-13

Round 2

Reviewer 3 Report

The authors answered the questions addressed in my review.

Believes the manuscript was sufficiently improved and justifies publication in Antibiotics.